# Small Molecules Promote the Rapid Generation of Dental Epithelial Cells from Human-Induced Pluripotent Stem Cells

**DOI:** 10.3390/ijms25084138

**Published:** 2024-04-09

**Authors:** Ximei Zhu, Yue Li, Qiannan Dong, Chunli Tian, Jing Gong, Xiaofan Bai, Jianping Ruan, Jianghong Gao

**Affiliations:** 1Key Laboratory of Shaanxi Province for Craniofacial Precision Medicine Research, College of Stomatology, Xi’an Jiaotong University, Xi’an 710004, China; zr19672@stu.xjtu.edu.cn (X.Z.); liyueeeeeeee@163.com (Y.L.); qianndong0915@stu.xjtu.edu.cn (Q.D.); 2Center of Oral Public Health, College of Stomatology, Xi’an Jiaotong University, Xi’an 710004, China; 18447062456@163.com; 3Department of Pediatric Dentistry, College of Stomatology, Xi’an Jiaotong University, Xi’an 710004, China; mysterygj@icloud.com (J.G.); bbjy33694@163.com (X.B.)

**Keywords:** human-induced pluripotent stem cells, dental epithelial, small molecules, spheroid

## Abstract

Human-induced pluripotent stem cells (hiPSCs) offer a promising source for generating dental epithelial (DE) cells. Whereas the existing differentiation protocols were time-consuming and relied heavily on growth factors, herein, we developed a three-step protocol to convert hiPSCs into DE cells in 8 days. In the first phase, hiPSCs were differentiated into non-neural ectoderm using SU5402 (an FGF signaling inhibitor). The second phase involved differentiating non-neural ectoderm into pan-placodal ectoderm and simultaneously inducing the formation of oral ectoderm (OE) using LDN193189 (a BMP signaling inhibitor) and purmorphamine (a SHH signaling activator). In the final phase, OE cells were differentiated into DE through the application of Purmorphamine, XAV939 (a WNT signaling inhibitor), and BMP4. qRT-PCR and immunostaining were performed to examine the expression of lineage-specific markers. ARS staining was performed to evaluate the formation of the mineralization nodule. The expression of PITX2, SP6, and AMBN, the emergence of mineralization nodules, and the enhanced expression of AMBN and AMELX in spheroid culture implied the generation of DE cells. This study delineates the developmental signaling pathways and uses small molecules to streamline the induction of hiPSCs into DE cells. Our findings present a simplified and quicker method for generating DE cells, contributing valuable insights for dental regeneration and dental disease research.

## 1. Introduction

The enamel, the outermost layer of the teeth, is the most susceptible to damage from trauma and bacteria [1]. Enamel is formed by the enamel organ, which consists of dental epithelial (DE) cells [2]. DE cells are lost upon tooth eruption. Given this challenge, transforming non-dental cells into DE cells emerges as a promising strategy for the clinical restoration of enamel and dental structures.

After the teeth emerge, DE, the cells responsible for enamel production, originate from the ectoderm layer. During the crucial phase of gastrulation, the ectoderm differentiates into two primary regions: the neural ectoderm (NE), which includes the neural plate and neural crest, and the non-neural ectoderm (NNE), encompassing pre-placodal ectoderm (PPE) and surface ectoderm (SE) [3]. The development of NNE requires not only induction signals but also the inhibition of mesendoderm-promoting genes [4]. BMP signaling has been demonstrated to be crucial for ectodermal patterning and the inhibition of NE fate [5,6]. While the TGFβ/Activin/Nodal pathway is known to induce mesoderm and endoderm fate [7]. Therefore, activating BMP signaling and inhibiting TGFβ signaling by BMP4 and SB431542 were used to induce NNE from human pluripotent stem cells [8]. It was reported that early retinoic acid (RA) application led to the emergence of unidentified non-neural cell fates [9]. A combination of RA and BMP4 was demonstrated to efficiently direct epithelial cell differentiation from human embryonic stem cells (hESCs) [10]. Nonetheless, these methods involving the activation of BMP signaling require growth factors to generate NNE. FGF signaling plays a pivotal role in maintaining the mesoderm [11] and inducing NE fate [12]. In the absence of FGF2-FGFR-ERK1/2 signaling, Activin/Nodal signaling alone is insufficient to drive mesendoderm and definitive endoderm formation [13]. Through inhibiting FGF signaling with SU5402, human-induced pluripotent stem cells (hiPSCs) have been successfully guided towards NNE fate, devoid of growth factors [14]. However, the application of SU5402 to the regeneration of DE remains unexplored.

The NNE region was further divided into PPE and SE [3]. The PPE contributes to the formation of both placodes and nearby epidermal areas, including the stomodeum, where OE originates [15,16]. For the PPE and OE to develop, BMP signaling inhibition is essential, alongside the activation of SHH signaling [17,18,19]. The suppression of BMP signaling has been applied to generate PPE appendages [20], including DE [21,22]. When BMP signaling inhibition is combined with SHH signaling activation, hESCs can efficiently generate OE [23].

To generate DE, or dental lamina, part of the oral epithelium in OE thickens within a specialized region [24,25]. Apart from its pivotal role in OE, SHH signaling determines the initiation location of DE [18,26]. Wnt-7b acts to repress SHH expression in oral epithelial cells, thus maintaining the boundaries between oral and DE cells [27]. In addition to SHH, BMP signaling also involves the proliferation and differentiation of DE [28,29]. Blocking these pathways in OE promotes more taste buds than DE differentiation [30]. Hence, the careful timing and regulation of these pathways are critical for accurately determining cell fate in the process of DE development.

For the purpose of advancing research in amelogenesis imperfecta and the regeneration of enamel, scientists have explored a variety of techniques to cultivate human DE cells in the lab. Induced pluripotent stem cells (iPSCs) are a potential and valuable cell resource to generate DE due to their patient-specific nature and non-immunogenic properties [31,32]. Current defined induction protocols, including induction with RA and BMP4, Noggin and EGF, BMP4 and EGF separately in three steps, and sequential stimulation with SAG (SHH signaling activator), BMP, etc. [21,22], have been developed to generate DE. Nevertheless, cytokines have some known drawbacks, including high cost, handling difficulty, immunogenicity, and a short half-life. Given the ability of small molecules to efficiently cross cell membranes, along with their affordability, stability, and ease of use [33], the focus has shifted towards establishing a defined differentiation procedure utilizing a precise combination of small molecules. This innovative approach, using small molecules to achieve stepwise FGF signaling inhibition, BMP signaling suppression combined with SHH signaling activation, and WNT signaling inhibition combined with both SHH and BMP signaling activation, aims to quickly produce DE cells from hiPSCs.

## 2. Results

### 2.1. NNE Cell Fate Determination through FGF Inhibition

We developed a defined protocol to generate DE from hiPSCs by stepwise manipulating the FGF, WNT, SHH, and BMP signaling pathways (Figure 1A). At the first stage, for the efficient large-scale production of NNE cells, we reviewed several existing differentiation protocols that utilize human pluripotent stem cells (hPSCs) and sorted out two methods, including using SB431542 and BMP4, applying RA and BMP4 as the positive controls in the phase of NNE induction [10,20]. Thus, three different induction protocols were evaluated to test whether FGF inhibition promotes the induction of NNE from hiPSCs: using 1 μM SB431542 (SB), an inhibitor of ALK4, ALK5, and ALK7, alongside 40 ng/mL BMP4; applying 1 μM RA, noting that the addition of BMP4 did not alter the NNE fate outcome (Appendix A); and administering 10 μM SU5402 (SU), a FGF receptor inhibitor (Figure 1B). During a 4-day analysis, we tracked the expression of lineage-specific marker genes (*DLX3*, *DLX5*, and *TFAP2A*) to monitor NNE differentiation. We found that all tested groups initiated NNE differentiation (Figure 1(Ca–d,Db,c), with SU markedly promoting NNE fate more than the others, as evidenced by a strong expression of TFAP2A (Figure 1(Db)) and a higher tendency of cells to develop into the PPE population (*SIX1*, PPE) over the SE (*K18*, SE) (Figure 1(Ce,i,Dd)). Additionally, SU appeared to repress the development of the NE (*SOX1*, NE), mesoderm (*BRACHYURY*, mesoderm), and endoderm lineages (*FOXA2*, endoderm) (Figure 1(Cf–h)). While RA hastened hiPSC differentiation and significantly increased *DLX5* expression compared to SU (Figure 1(Ca–d)), it skewed the NNE population towards becoming SE, as depicted (Figure 1(Ce)). Upon the induction of NNE, the disappearance of colony shape and dense arrangement implied the differentiation of hiPSCs (Figure 1(Da)).

### 2.2. Cell Density and the Duration of Differentiation Are Important for NNE Differentiation

Cell density has a substantial impact on the differentiation outcomes of the three germ layers in embryonic stem cells [23]. Thus, the potential impact of cell density on the production of NNE cells by adjusting the initial seeding density was investigated (Figure 2A). At the lowest cell density (3.5 × 10^4^ cells/cm^2^), there was minimal expression of *DLX3*, *DLX5*, and *TFAP2A*. With an increase in cell density, the expression levels of these markers improved, reaching optimal levels at a density of 14 × 10^4^ cells/cm^2^ (Figure 2(Ba–c,C)). The expression of *BRACHYURY* diminished, whereas *SOX1* and *FOXA2* exhibited no discernible changes with increasing cell density (Figure 2(Bd–f)). It suggested that increasing cell density promoted NNE while inhibiting mesoderm differentiation through the inhibition of FGF signaling.

We investigated the impact of FGF inhibition duration on NNE differentiation, finding that the levels of DLX3, DLX5, and TFAP2A significantly rose by day 4. DLX3 and TFAP2A remained highly expressed, while DLX5’s expression diminished by day 10 (Figure 2(Da–c,Ea)). Additionally, SIX1 was detectable on day 4, but its expression decreased as the induction period extended (Figure 2(Dd,Eb)). The establishment of NNE is not solely dependent on TFAP2A expression [14]; therefore, despite the sustained increase in TFAP2A during the induction, the expression of DLX3, DLX5, and SIX1 suggested that a 4-day differentiation period might present an appropriate developmental window for NNE patterning. Therefore, a seeding density of 14 × 10^4^ cells/cm^2^ with a 4-day treatment might be efficient for the establishment of NNE fate.

### 2.3. PPE Specification and the Emergence of OE through BMP Inhibition and SHH Activation

During the late blastula stage, activating BMP signaling prevents the development of the PPE [34]. Activating SHH signaling during the PPE stage enhances the progression towards OE development [35]. To delineate whether BMP signaling inhibition and SHH signaling activation are contributing to PPE formation, LDN193189 (LDN), a BMP type I receptor inhibitor, and Purmorphamine (Purm), a SHH receptor activator, were applied to inhibit BMP and activate SHH signaling, respectively (Figure 3A). The results showed that LDN significantly increased the expression of PPE markers SIX1 and EYA1 (Figure 3(Ba,b)). When combined with Purm, the expression of these genes was further enhanced, although Purm alone was not beneficial to SIX1 expression (Figure 3(Ba,b,C)).

During the induction process of PPE, the expression of OE markers of PITX2 and PITX1 peaked on day 6 (Figure 3(Dc,d,Eb,c)) and declined as the induction continued, a trend that did not precisely correlate with the pattern of *SIX1* and *EYA1* expression, which continued to rise till day 8 (Figure 3(Da,b,Ea)). These findings indicated that the OE fate started to occur prior to the complete establishment of the PPE region.

### 2.4. DE Formation through SHH and BMP Activation, Alongside with WNT Inhibition

The SHH, BMP, and WNT signaling pathways, which are evolutionarily conserved across species, play crucial roles in regulating tooth development [36]. Throughout tooth formation, a variety of WNTs and their pathway intermediates are expressed. Specifically, Wnt4 and Wnt6 are found in both the outer OE and DE, whereas Wnt3 and Wnt7b are not expressed in the DE [37,38]. WNT signaling suppression of PITX expression could clarify why WNT/β-catenin signaling remains inactive in the epithelial stem cells or stellate reticulum of the labial cervical loop [24,39]. To gain insight into the role of WNT signaling in the regulation of DE differentiation, the effects of CHIR99021 (CHIR), a WNT signaling activator, and XAV939 (XAV), a WNT signaling inhibitor, were compared. The expression of ameloblast markers *SP6* and *AMBN* increased on CHIR treatment (Figure 4(Cb,c)), as previously reported in the literature, while the expression of *PITX2* was notably inhibited (Figure 4(Ca)). Conversely, the addition of XAV enhanced the expression of *PITX2*, *SP6*, and *AMBN* (Figure 4(Ca–c)), and the concomitant addition of Purm and BMP4 further improved the effect (Figure 4(Cd–f,D)). During the induction of DE, the expression of PITX2, AMBN, and SP6 peaked on day 8 and decreased thereafter (Figure 4E,F), confirming that a total treatment of 8 days may be sufficient for the acquisition of DE fate.

The induced cells enlarged and displayed polygonal shapes in the final induction phase reminiscent of mouse DE (mDE) (Figure 4B), with strong labeling of SOX2 and Ki67, indicating DE properties along with stemness and proliferative capabilities (Figure 4G). To test the functionality of hiPSC-derived DE, it was further differentiated using an ameloblast differentiation medium. The emergence of mineralization nodules indicated the mineralisation capacity -like ameloblasts (Figure 4(Ha)). Additionally, when co-cultured with hiPSC-derived DE using a transwell system, human dental pulp stem cells (hDPSCs) exhibited odontogenic differentiation (Figure 4(Hb)). These observations suggested that a combination of WNT signaling inhibition, SHH, and BMP signaling activation might be effective for driving DE formation (Figure 4A).

### 2.5. DE in Spheroid Culture and Assembled with hDPSCs Allows Further Differentiation

To mimic the in vivo microenvironment of DE, the hiPSC-derived DE were cultured in suspension, enabling them to self-organize into spheroids (Figure 5A). These spheroids displayed an irregular sphere morphology, featuring an outer layer of stratified cuboidal epithelium (CE) and a loose inner part (Figure 5(Ba,b)). HiPSC-derived DE were immunopositively stained for CK14, indicating the epithelial phenotype of these culture cells. Expression of AMBN and AMELX was predominantly observed in the outer layer, with sporadic expression within the spheroid’s inner regions. Notably, within the inner areas where cells were arranged around voids, AMBN expression was oriented towards these gaps (Figure 5(Bc–e)). To further assess the potential function of our hiPSC-derived DE cells, the epithelial-mesenchymal interaction was investigated in vitro (Figure 5(Ca)). In the assembloids, the hiPSC-derived DE was located in the outer part, displaying a much looser arrangement compared with the inner part, where hDPSCs were located (Figure 5(Cb)). The assembloids comprised distinct epithelial and mesenchymal (CD105+) domains (Figure 5(Cc)). In addition to AMBN and AMELX, DSPP was robustly expressed in hiPSC-derived DE and the out layer where hDPSCs faced toward hiPSC-derived DE (Figure 5(Cd,e)). DSPP is transiently expressed in early ameloblasts and is associated with ameloblast differentiation and enamel biomineralization [40]. The results above indicated that the induced cells underwent ameloblast differentiation in spheroid and assembloids cultures.

## 3. Discussion

The presently established protocols for DE differentiation rely heavily on cytokines, while our approach majorly utilizes small molecules and a select growth factor to produce hiPSC-derived DE. The reduced use of cytokines minimized both cost and instability. Meanwhile, small molecules enhance the overall efficiency of the differentiation process by expediting the cellular response to stimuli. Our research presented a comprehensive analysis and optimization of protocols for the differentiation of hiPSCs into NNE and subsequently into DE, culminating in the formation of structures resembling the dental epithelium in spheroid culture. This work is vital in addressing the challenges associated with enamel regeneration and amelogenesis imperfecta.

The determination of NNE cell fate through the inhibition of FGF signaling highlighted the critical role of pathway modulation in NNE differentiation. Our comparison of various induction protocols underscored the importance of precise signaling manipulation, with the use of SU (FGF signaling inhibitor) emerging as a particularly effective method for promoting NNE differentiation. A report has stated that dual SMAD inhibition by adding SB (TGFβ signaling inhibitor) and LDN (BMP signaling inhibitor) prior to SU treatment is capable of preventing mesoderm and endoderm differentiation, thereby favoring the acquisition of the NNE fate [14]. However, in our experiments, when SB and LDN were introduced at the onset of differentiation, the expression of *DLX3*, *DLX5*, and *TFAP2A* was significantly suppressed, while mesoderm and endoderm differentiation were not inhibited (refer to Appendix A). This observation could potentially be attributed to the complete abrogation of endogenous BMP signaling by these factors, as BMP signaling has been demonstrated to be crucial for the formation of NNE [41]. SU has not been used in the regeneration of DE. In our study, we found that SU alone markedly potentiates the NNE fate, thereby offering valuable insights for future explorations into the area of early human ectoderm cell type development.

Cell density and differentiation duration were identified as crucial parameters influencing NNE differentiation [42]. Our investigations revealed that an increasing cell density actively facilitates NNE differentiation. Prolonged exposure to SU effectively suppressed SIX1 while maintaining the expression of TFAP2A, as evident in our results. Consequently, a 4-day induction period was sufficient for the acquisition of NNE fate, paving the way for the progression to the subsequent stage. This observation reinforces existing literature on the impact of cell density on stem cell fate decisions and introduces a nuanced understanding of how these variables can be optimized to enhance the efficiency of DE differentiation.

Our exploration into PPE specification and the emergence of OE through the modulation of BMP and SHH signaling pathways further elucidated the complex interplay of developmental signals in DE differentiation. LDN upregulated the expression of SIX1 and EYA1, suggesting that BMP inhibition played a crucial role in the acquisition of PPE, in alignment with the findings reported by Litsiou et al. [35]. This implied that BMP worked within a narrow window to induce the PPE. When combined with SHH activation, the PPE’s fate was strengthened. LDN and Purm (an SHH signaling activator) efficiently promoted the emergence of OE at the same time, indicating BMP inhibition combined with SHH activation could also efficiently direct the differentiation of OE from hiPSCs, which was consistent with the report by Leung et al. [23]. In Leung’s study, the addition of Purm following a two-day treatment with LDN and Purm further increased the expression of PITX2. This observation differed from our findings, potentially due to disparities in the cellular lineage utilized. In our study, the fate of OE emerged during the induction of PPE, as evidenced by the strong expression of PITX1 and PITX2 during the PPE induction period. These findings not only contribute to our fundamental understanding of tooth development but also offer potential avenues for the regeneration of specific dental tissues.

The investigation into DE formation through WNT regulation offers a novel perspective on the regulation of DE differentiation. CHIR (a WNT signaling activator) stimulated the upregulation of *SP6* and *AMBN* expression but inhibited the expression of *PITX2*. Presumably, the activation of WNT signaling promoted the differentiation of preexisting DE into ameloblasts while suppressing DE differentiation. Conversely, XAV (a WNT signaling inhibitor) increased the expression of PITX2, thereby suggesting that the suppression of WNT signaling facilitated the formation of DE, given the absence of WNT signaling in the DE region, as reported by Soukup and Suomalainen et al. [24,39]. The combination of XAV with either Purm or BMP4 did not result in a further elevation of PITX2 expression. However, when XAV was combined with both Purm and BMP, the expression of PITX2 was further augmented. This result might be attributed to the reciprocal regulations between the WNT, BMP, and SHH signaling pathways, underscoring the intricate and context-dependent nature of each pathway within its respective network [29,43,44]. Through meticulous modulation of FGF, BMP, SHH, and WNT signaling via small molecules and one solitary growth factor, we successfully regenerated hiPSC-derived DE-functioning amelogenesis and odontogenesis within 8 days, significantly shortening the induction period in comparison to the 12- or 16-day duration [21,22]. This aspect of our research highlighted the potential of targeted signaling pathway modulation to overcome the limitations of current differentiation protocols. However, due to the inherent limitation of epithelial sheets being unsuitable for continuous and long-term in vitro culture, we encountered challenges in achieving stable expansion of our induced DE cells in monolayer culture. Furthermore, there is a lack of reports on the successful expansion of hiPSC-derived DE cells in monolayer culture [21,22]. Further research can focus on modifying the culture medium and selecting an appropriate basement membrane matrix to enhance the growth potential of hiPSC-derived DE cells. Additionally, spheroid culture may serve as an alternative approach for long-term expansion of hiPSC-derived DE cells, as the findings of our study, along with those of previous studies [45,46], have consistently demonstrated this phenomenon.

The use of spheroid culture to mimic the in vivo microenvironment of DE represents a significant advancement in dental tissue engineering. The hiPSC-derived DE spheroid exhibited robust labeling for AMBN and AMELX, thereby demonstrating its potential to initiate ameloblast differentiation when cultured in a spheroid. This observation aligned with the findings reported by Kim et al. [47]. It has been reported that the temporal structure affects the secretion of extracellular matrix (ECM) from mesenchymal stem cells and promotes the secretion of ECM by OE cells [47,48]. We speculated that the suspension culture might promote secretion of the enamel matrix protein of hiPSC-derived DE. Our findings demonstrated that spheroid culture promoted a more pronounced differentiation of hiPSC-derived DE, offering a promising approach to creating three-dimensional dental tissues with enhanced structural and functional complexity. This methodological innovation opens new pathways for the development of regenerative therapies and the study of dental diseases.

The enamel organ comprises a small cell population, which includes SR, SI, OEE, and IEE, where ameloblasts originate [49]. In addition to signaling, including WNT, TGF-β, etc. [50], cell-cell interaction and dentine matrix are all requirements for ameloblast cytodifferentiation [49]. Further research is needed to expand and differentiate DE into diverse enamel organ cells.

## 4. Materials and Methods

### 4.1. Cell Lines and Culture Conditions

hiPSCs (hiPSCs-B1 and hiPSCs-F1 were generated from peripheral blood mononuclear cells and dermal fibroblasts) (Cellapybio, Beijing, China) were used in this study. hiPSCs-B1 was the main cell line used in the DE differentiation study, and the DE differentiation protocol was confirmed by hiPSCs-F1. Both cells were seeded on 6-well plates coated with growth factor-reduced Matrigel (Corning, Bedford, MA, USA) and cultured in mTeSR1 stem cell medium (STEMCELL Technologies Inc., Cambridge, MA, USA). The cells were maintained at 37 °C in a humidified atmosphere containing 5% CO_2_ with medium changes daily. Cells were passaged by Versene (Gibco, Carlsbad, CA, USA) when cells reached 80% to 90% confluency.

### 4.2. Dental Epithelial (DE) Cell Differentiation in Monolayer

The hiPSCs were maintained as specified until they attained 80% confluence. Following this, they were transferred to differentiation medium I, which was composed of DMEM/F12 supplemented with 15% (*v*/*v*) knockout serum replacement (KSR), 1% GlutaMax I, 1% nonessential amino acid stock, 1% Penicillin/Streptomycin (all from Gibco), and 0.1 mM β-mercaptoethanol (Sigma-Aldrich, St. Louis, MO, USA). From Day 0 to Day 4, the cells were cultured in differentiation medium I with 10 μM SU5402. From Day 4 to Day 6, the medium was changed to differentiation medium II, which was a 1:1 mixture of differentiation medium I and keratinocyte serum-free medium (ScienCell Research Laboratories, Carlsbad, CA, USA) (1:1), enriched with 1 μM LDN193189 and 2 μM purmorphamine. From Day 6 to Day 8, the medium was changed to differentiation medium III, consisting of keratinocyte serum-free medium with 1 μM XAV939, 2 μM purmorphamine, and 40 ng/mL BMP4 (Peprotech, Rocky Hill, NJ, USA). All small molecules were purchased from MCE (MedChem Express, Monmouth Junction, NJ, USA).

### 4.3. RNA Extraction and Quantitative Real-Time PCR (qRT-PCR) Analysis

Total RNA was extracted using Trizol (TaKaRa, Tokyo, Japan). 2 μg of RNA was reverse transcribed using an RT-PCR kit (Yeasen, Shanghai, China) according to the manufacturer’s protocol. qRT-PCR was performed using Hieff^®^ qPCR SYBR Green Master Mix (Yeasen) on the CFX96 Real-Time PCR Detection System (Bio-Rad, Hercules, CA, USA), and the data were normalized to GAPDH expression. The primers used were listed in Appendix A (Sangon, Shanghai, China).

### 4.4. Immunostaining

The cells were fixed in 4% paraformaldehyde (PFA) (Boster, Wuhan, China) for 15 min, washed with Phosphate-Buffered Saline (PBS) (Boster), and permeabilized in 0.1% Triton X-100/PBS (Solarbio, Beijing, China) for 10 min. After blocking with 3% Bovine Serum Albuminn (BSA)/PBS (Boster) for 1 h, the cells were incubated with primary antibodies overnight at 4 °C. Subsequently, the cells were washed in PBS and incubated with the secondary antibodies for 1 h before being washed three times with PBS. Antifade Mounting Medium with DAPI (Beyotime, Shanghai, China) was used for nuclei staining. The cells were photographed by Laser Scanning Confocal Microscopy (Olympus FV3000; Olympus, Tokyo, Japan). The primary antibodies used were listed in Appendix A.

### 4.5. Development of hiPSCs Derived DE Cells and Co-Culturing hiPSCs Derived DE with hDPSCs in Sphere Culture

On day 8, differentiated hiPSC-derived DE cells were trypsinized using accutase (Yeasen). 1 × 10^5^ cells were re-plated in pluronic F-127 (Beyotime)-treated non-tissue culture-treated 96-well plates containing Keratinocyte serum-free medium with 10 μM Y-27632 (MCE). After centrifugation for 1 min at 300g and 4 °C, the sphere cultures were maintained at 37 °C in 5% CO_2_ for two weeks, and the medium was changed every 3 days until further analysis.

To reconstitute DE and hDPSCs, hDPSCs were trypsinized using 0.25% trypsin (Gibco). Firstly, hDPSCs (5 × 10^4^ cells) were sedimented by centrifugation (300 g for 1 min at 4 °C), followed by deposition of hiPSC-derived DE cells (1 × 10^5^ cells at 300 g for 2 min at 4 °C) [46]. The cells were cultured in a 1:1 mixture of medium containing α-MEM (Gibco), 10% FBS (Gibco), 1% Penicillin/Streptomycin (Gibco), and keratinocyte serum-free medium for two weeks. The assembloids were later sampled for further analysis.

### 4.6. Cryosectioning and Immunostaining for the Sphere

The spheroids were fixed in 4% PFA overnight at 4 °C. After being rinsed three times with PBS for 5 min each, the spheroids were embedded in Tissue-Tek^®^ OCT compound (Sakura Finetek USA, Inc., Torrance, CA, USA) and slowly frozen on a metal block chilled on dry ice. The frozen spheroids were cut using a Cryostat (Leica CM1950; Leica, Wetzlar, Germany) to create 10mm slices and fixed on adhesion glass slides (Citotest, Haimen, China) for staining. The spheroidal sections were fixed in 4% PFA for 10–15 min at room temperature and later washed three times with PBS for 5 min each. Slides were then immersed in 0.2% Triton X-100 at room temperature for 10 min to facilitate permeabilization. Later, blocking was performed for 1 h at room temperature in a humidified chamber with a blocking buffer consisting of 3% BSA/PBS. The spheroids were incubated with primary antibodies overnight at 4 °C in a humidified chamber. After being washed in PBS, the slides were transferred to a humidified chamber with secondary antibodies. Secondary antibodies were applied for 1 h at room temperature in the same blocking agent, followed by rinsing the slides with PBS. Antifade Mounting Medium with DAPI was used for nuclei staining. Slides were stored at 4 °C for imaging. The slides were photographed by Laser Scanning Confocal Microscopy (Olympus FV3000).

### 4.7. Alizarin Red Staining

The spheroidal sections were fixed in 4% PFA at room temperature for 10 min. Rinse the section with deionized distilled water three times for 5 min each. Sections were incubated in 2% Alizarin red S solution (pH4.2) (Beyotime) at room temperature for 30 min. The section was rinsed with several changes of deionized distilled water for 5 min and then photographed by a microscope (Olympus FSX100).

### 4.8. Statistical Analysis

Student’s T-test and one-way ANOVA were used for statistical analysis, as indicated for each data set. The results are presented as mean ± SEM, and *p* value <0.05 was used as the criterion for statistical significance. Each dataset was generated from at least three independent experiments. Additionally, the levels of significance were denoted as * *p* < 0.05, ** *p* < 0.01, *** *p* < 0.001, **** *p* < 0.0001, and n.s., not significant, with the data analyzed using GraphPad Prism software (version 8.0; GraphPad Software Inc., San Diego, CA, USA).

## 5. Conclusions

In summary, our study provides a comprehensive framework for the differentiation of hiPSCs into DE tissues, emphasizing the importance of signaling pathway modulation, cell density, and culture conditions in optimizing stem cell differentiation protocols. Through meticulous modulation of FGF, BMP, SHH, and WNT signaling via small molecules and one solitary growth factor to establish hiPSCs-derived DE functioning amelogenesis and odontogenesis within 8 days. The insights gained from this research offer new strategies for the treatment of dental diseases and the regeneration of dental tissues. Future work will need to focus on refining these differentiation protocols further, exploring the in vivo potential of hiPSC-derived DE cells, and expanding our understanding of the molecular mechanisms underlying dental tissue regeneration.

## Figures and Tables

**Figure 1 ijms-25-04138-f001:**
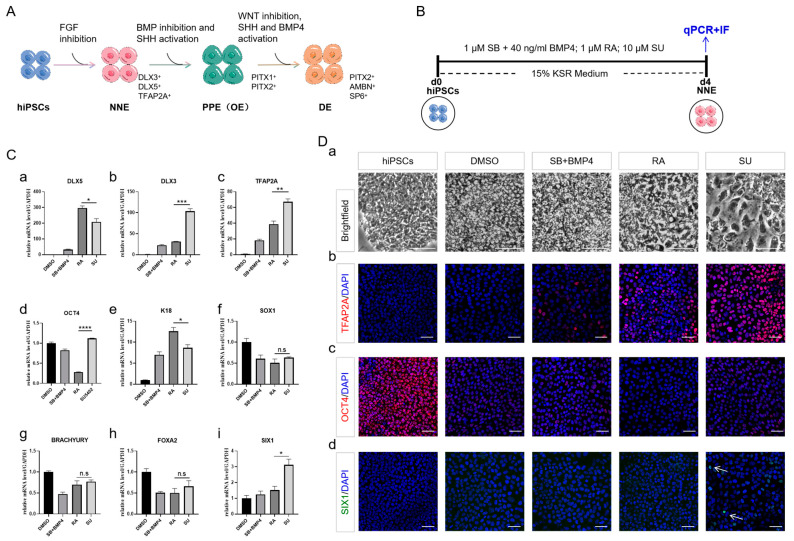
FGF inhibition promotes the induction of NNE. (**A**) Schematic flow chart showing the differentiation protocol targeting signaling pathways. (**B**) Schematic flow chart showing the three NNE differentiation protocols. (**C**) qRT-PCR analysis showing expression of markers for different germ layers on day 4 of differentiation. NNE markers *DLX5*, *DLX3*, and *TFAP2A* (**a**–**c**); Stemness marker *OCT4* (**d**); SE marker *K18* (**e**); NE marker *SOX1* (**f**); Mesoderm marker *BRACHYURY* (**g**); Endoderm marker *FOXA2* (**h**); PPE marker *SIX1* (**i**). (**D**) Brightfield indicated the cell morphology and immunostaining of markers for NNE, stemness and PPE on day 4 of differentiation. Brightfield indicated the change in cell morphology (**a**); Immunostaining of NNE marker of TFAP2A (red), Stemness marker OCT4 (red), and PPE marker SIX1 (green) (**b**–**d**). Arrows indicate SIX1^+^ cells. Scale bar, 50 µm. Human-induced pluripotent stem cells (hiPSCs). Non-neural ectoderm (NNE). Surface ectoderm (SE). Neural ectoderm (NE). Pre-placodal ectoderm (PPE). SB431542 (SB). SU5402 (SU). Knockout serum replacement (KSR). The data shown are the mean ± SEM of three independent experiments (three technical replicates per biological replicate). * *p* < 0.05, ** *p* < 0.01, *** *p* < 0.001, **** *p* < 0.0001, and n.s., not significant.

**Figure 2 ijms-25-04138-f002:**
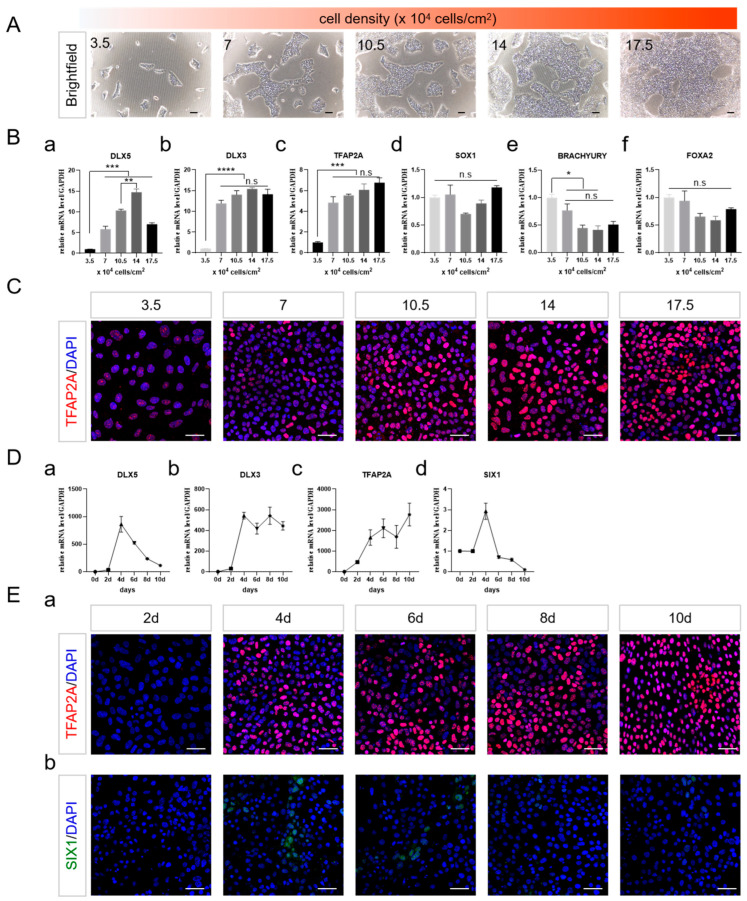
NNE Optimization of cell density and duration for NNE induction. (**A**) Brightfield indicated the change in cell density. Scale bar, 100 µm. (**B**) qRT-PCR analysis showing expression of markers for different germ layers on day 4 of differentiation. NNE markers *DLX5, DLX3*, and *TFAP2A* (**a**–**c**); NE marker *SOX1* (**d**); Mesoderm marker *BRACHYURY* (**e**); Endoderm marker *FOXA2* (**f**). (**C**) Immunostaining of TFAP2A (red) on day 4 of differentiation. Scale bar, 50 µm. (**D**) qRT-PCR analysis showing expression of NNE and PPE makers with prolonged induction. NNE markers *DLX5*, *DLX3, TFAP2A* (**a**–**c**); PPE maker *SIX1* (**d**). (**E**). Immunostaining of NNE and PPE markers on day 4 of differentiation. NNE marker TFAP2A (red) (**a**); PPE marker SIX1 (green) (**b**). Scale bar, 50 µm. The data shown are the mean ± SEM of three independent experiments (three technical replicates per biological replicate). * *p* < 0.05, ** *p* < 0.01, *** *p* < 0.001, **** *p* < 0.0001, and n.s., not significant.

**Figure 3 ijms-25-04138-f003:**
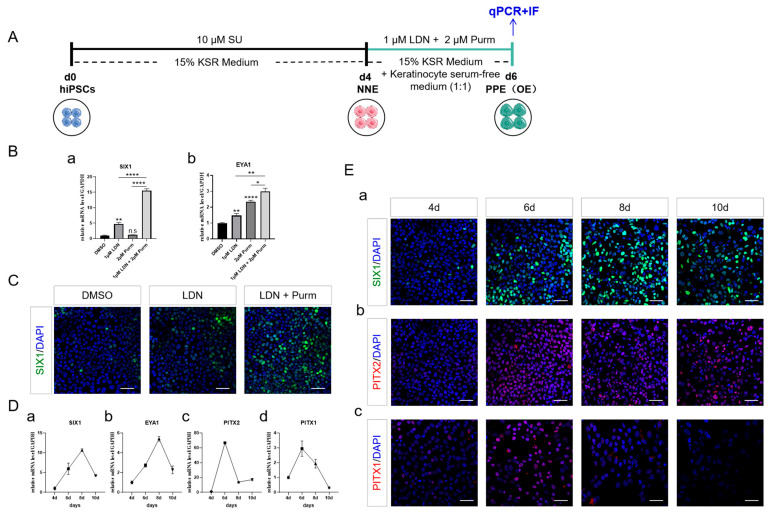
Optimization conditions for PPE formation. (**A**) Schematic representation of the PPE differentiation protocol. (**B**) qRT-PCR analysis showing expression of PPE markers by modulating BMP and SHH signaling. *SIX1* (**a**); *EYA1* (**b**). The statistical analysis was performed by comparing the DMSO control. (**C**) Immunostaining of SIX1 (green) on day 6 of differentiation. Scale bar, 50 µm. (**D**) qRT-PCR analysis showing expression of PPE and OE markers with prolonged induction. *SIX1* (**a**); *EYA1* (**b**)*; PITX2* (**c**); *PITX1* (**d**). (**E**) Immunostaining of PPE and OE markers with prolonged induction. SIX1 (green) (**a**); PITX2 (red) (**b**); PITX1 (red) (**c**). Scale bar, 50 µm. Oral ectoderm (OE). LDN193189 (LDN). Purmorphamine (Purm). The data shown are the mean ± SEM of three independent experiments (three technical replicates per biological replicate). * *p* < 0.05, ** *p* < 0.01, **** *p* < 0.0001, and n.s., not significant.

**Figure 4 ijms-25-04138-f004:**
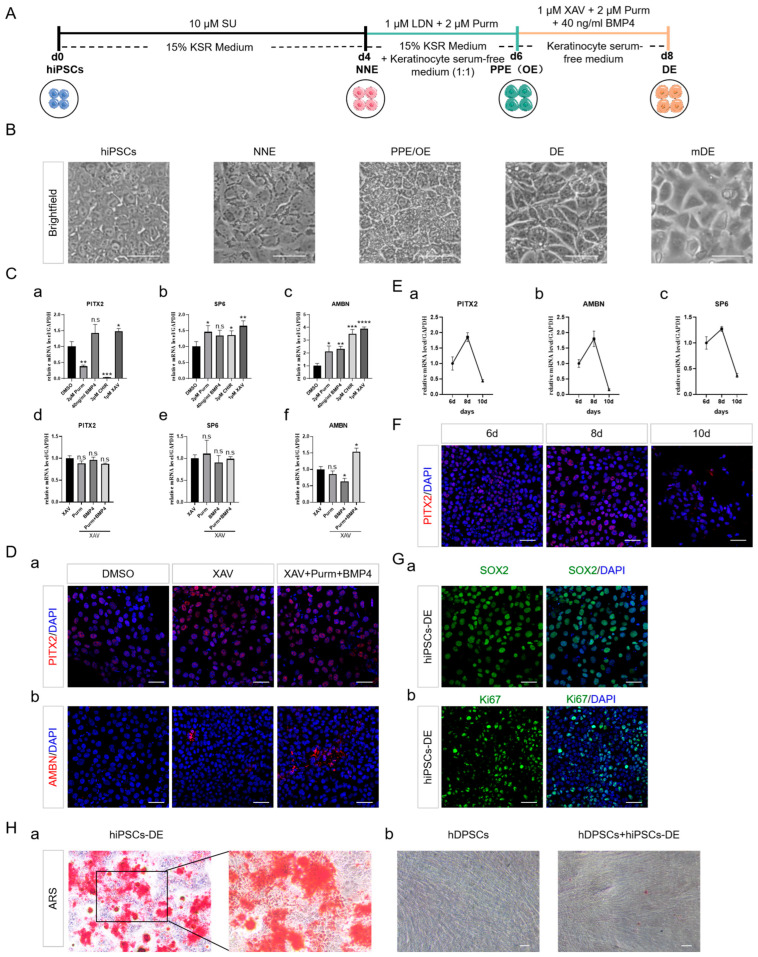
Optimization conditions for DE formation. (**A**) Schematic representation of the DE differentiation protocol. (**B**) Cell morphology changes in the differentiation process. Scale bar, 50 µm. (**C**) qRT-PCR analysis showing expression of DE and ameloblast markers by modulating WNT, SHH, and BMP signaling. DE marker *PITX2*(**a**,**d**); Ameloblast markers *AMBN* and *SP6* (**b**,**c**,**e**,**f**). The results were normalized to the DMSO control (**a**–**c**) and the XAV control (**d**–**f**). (**D**) Immunostaining of DE and ameloblast markers on day 8 of differentiation. PITX2 (red) (**a**); AMBN (red) (**b**). Scale bar, 50 µm. (**E**) qRT-PCR analysis showing the expression of DE and ameloblast markers with time. *PITX2* (**a**); *AMBN* (**b**); *SP6* (**c**). (**F**) Immunostaining of PITX2 (red) with time. Scale bar, 50 µm. (**G**) Immunostaining of DE and proliferation markers on day 8 of differentiation. DE marker SOX2 (green) (**a**); Proliferation marker Ki67 (green) (**b**). Scale bar, 50 µm. (**H**) ARS staining of hiPSC-derived DE and hDPSCs after inductionARS staining of hiPSC-derived DE after induction of ameloblast differentiation medium (**a**); ARS staining of hDPSCs after coculturing with hiPSCs derived DE by the transwell system (**b**). Scale bar, 100 µm. Dental epithelial (DE). Mouse dental epithelial (mDE). hiPSCs-derived DE (hiPSCs-DE). Human dental pulp stem cells (hDPSCs). CHIR99021 (CHIR). AV939 (XAV). The data shown are the mean ± SEM of three independent experiments (three technical replicates per biological replicate). * *p* < 0.05, ** *p* < 0.01, *** *p* < 0.001, **** *p* < 0.0001, and n.s., not significant.

**Figure 5 ijms-25-04138-f005:**
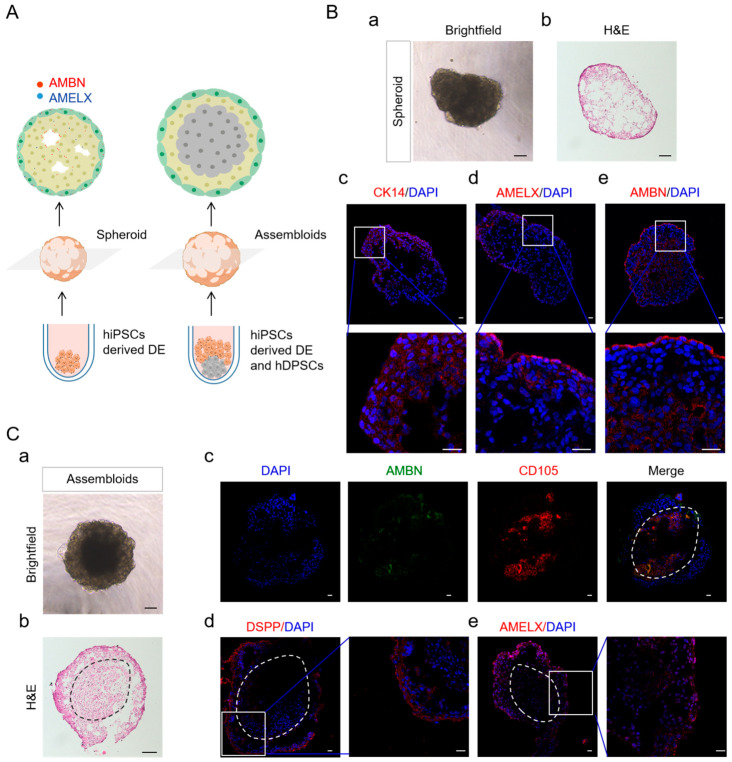
Characters of hiPSCs-derived DE and the assembloids of hiPSCs-derived DE and hDPSCs in spheroid culture. (**A**) Schematic representation of the formation of hiPSCs-derived DE spheroid and the assembloids of hiPSCs-derived DE and hDPSCs. (**B**) hiPSC-derived DE was cultured in keratinocyte serum-free medium using a U-bottom plate to form a spheroid. Scale bar, 100 µm (**a**); hematoxylin and eosin staining of the spheriod after 2 weeks. Scale bar, 100 µm (**b**); Immunostaining of the ameloblast markers AMBN, AMELX, and CK14 in spheriod after 2 weeks. Scale bar, 50 µm (**c**–**e**). (**C**) hiPSC-derived DE and hDPSCs were cocultured in keratinocyte serum-free medium and odontogenic medium (1:1) using a U-bottom plate to form assembloids. Scale bar, 100 µm (**a**); Hematoxylin and eosin staining of the spheriod after 2 weeks. Scale bar, 100 µm (**b**); Immunostaining of AMBN, CD105, DSPP, and AMELX in spheriod after 2 weeks. Scale bar, 50 µm (**c**–**e**). The dotted area demarcates the mesenchymal cells.

## Data Availability

The data that support the findings of this study are available from the corresponding author upon reasonable request and within its Appendix A published online.

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
