# Peer review of "Small Molecules Promote the Rapid Generation of Dental Epithelial Cells from Human-Induced Pluripotent Stem Cells"

_ijms, 2024, doi:10.3390/ijms25084138_

Round 1
Reviewer 1 Report
Comments and Suggestions for Authors
I read the manuscript with interest and I want to congratulate you for the work you have done. The study is very interesting and brings innovative elements to this research direction.The research is well conducted, the results are presented in a clear manner .
I would still have a few observations:
1. I understood that the study represents an element of progress in the mentioned research direction, but I am sure that it still has limits. Please specify them.
2. The use of product or company names in articles is not indicated.
3. . Are you thinking of continuing research in vivo?
Author Response
Dear Reviewer:
Thank you very much for your comments and professional advice. Those comments are all valuable and very helpful for revising and improving our paper, as well as the important guiding significance to our research. We have made corrections which we hope meet with approval. In this revised version, changes to our manuscript were all highlighted within the document by using yellow-colored text. Furthermore, we would like to show the details as follows:
- I understood that the study represents an element of progress in the mentioned research direction, but I am sure that it still has limits. Please specify them.
Reply: The limitation of the study has been added in the fifth paragraph on the Discussion section. Revised portions are as follows:
However, due to the inherent limitation of epithelial sheets being unsuitable for continuous and long-term in vitro culture, we encountered challenges in achieving stable expansion of our induced DE cells in monolayer culture. Furthermore, there is a lack of reports on the successful expansion of hiPSCs-derived DE cells in monolayer culture [21, 22]. Further researches can focus on modifying the culture medium and selecting an appropriate basement membrane matrix to enhance the growth potential of hiPSCs-derived DE cells. Additionally, spheroid culture may serve as an alternative approach for long-term expansion of hiPSCs-derived DE cells, as the findings of our study, along with those of previous studies [34, 46], have consistently demonstrated this phenomenon.
- The use of product or company names in articles is not indicated.
Reply: All of the product or company names have been added to the Materials and Methods sections.
- Are you thinking of continuing research in vivo?
Reply: Yes. The research holds great significance in providing a cell resource for enamel regeneration. In order to demonstrate the potential of hiPSCs-derived DE cells in enamel formation, it is imperative to conduct in vivo studies. In our research, expression of amelogenin, the main enamel matrix protein, within a three-dimensional culture system demonstrates the formation of functional ameloblast, underscoring the crucial role played by spatial environment in promoting ameloblast differentiation. Therefore, we are trying to find an appropriate scaffold that can effectively support and guide the differentiation of hiPSCs-derived dental epithelial cells to facilitate proper orientation for secretion of enamel matrix in vivo.
Thank you again for your positive comments and valuable suggestions to improve the quality of our manuscript.
Yours sincerely,
Ximei, Zhu
Reviewer 2 Report
Comments and Suggestions for Authors
In this paper authors developed a three-step protocol to convert hiPSCs into DE cells in 8 days. All phases are extensively described. The paper is well organized and provides some new and interesting data. For this reason, I propose to accept it for publication. In following you can find some minor importance comments.
It is not clear what the x-axes in figures 2B and D represents. The same is also happened in many other figures. All these should be corrected.
It is not clear why optimal levels at a density of 14 × 104 cells/cm2 are reached. Fig. 2Bc gives an other values as optimum.
Comments on the Quality of English LanguageMinor editing of English language required
Author Response
Dear Reviewer:
Thank you very much for your comments and professional advice. Those comments are all valuable and very helpful for revising and improving our paper, as well as the important guiding significance to our researches. We have made correction which we hope meet with approval. In this revised version, changes to our manuscript were all highlighted within the document by using yellow-colored text. Furthermore, we would like to show the details as follow:
- It is not clear what the x-axes in figures 2B and D represents. The same is also happened in many other figures. All these should be corrected.
Reply: The titles of X-axes have been added in figures 2B, 2D, 3D and 4E.
- It is not clear why optimal levels at a density of 14 × 104 cells/cm2 are reached. Fig. 2Bc gives an other values as optimum.
Reply: The mRNA expression levels of DLX3 and TFAP2A remained unchanged across cell densities ranging from 7 to 14 × 104 cells/cm2 (Figure 2Bb and Bc), indicating that all tested cell densities within this range were suitable for non-neural ectoderm induction. However, the mRNA expression level of DLX5 was significantly higher compared to other groups. Considering the overall results, we propose that a cell density of 14 × 104 cells/cm2 may be more efficient for inducing non-neural ectoderm.
- Minor editing of English language required
Thanks for your suggestion. We tried our best to improve the manuscript and made some changes to the manuscript. These changes will not influence the content and framework of the paper. And here we did not list the changes but marked in yellow in the revised paper.
Thank you again for your positive comments and valuable suggestions to improve the quality of our manuscript.
Yours sincerely,
Ximei, Zhu
Reviewer 3 Report
Comments and Suggestions for Authors
Very interesting work! The topic is extremely important in terms of therapeutic problems in patients, for example, with amelogenisis imperfecta, which the authors emphasized in the conclusions and introduction. Both the material and methods as well as the results were presented very clearly. The discussion is sensibly led, only it would be worth considering updating the literature references (more than half are older than 10 years). Congratulations on the research!
Author Response
Dear Reviewer:
Thank you very much for your comments and professional advice. Those comments are all valuable and very helpful for revising and improving our paper. We have made correction which we hope meet with approval. In this revised version, changes to our manuscript were all highlighted within the document by using yellow-colored text. Furthermore, we would like to show the details as follow:
- It would be worth considering updating the literature references (more than half are older than 10 years).
Reply: We sincerely appreciate the valuable comments. We have checked the literature carefully and updated 19 references that are older than 10 years to the last 5 years in the revised manuscript.
Thank you again for your positive comments and valuable suggestions to improve the quality of our manuscript.
Yours sincerely,
Ximei, Zhu